# It’s Time to Replace the Term “Heavy Metals” with “Potentially Toxic Elements” When Reporting Environmental Research

**DOI:** 10.3390/ijerph16224446

**Published:** 2019-11-13

**Authors:** Olivier Pourret, Andrew Hursthouse

**Affiliations:** 1UniLaSalle, AGHYLE, 19 rue Pierre Waguet, 60000 Beauvais, France; 2School of Computing, Engineering & Physical Sciences, University of the West of Scotland, Paisley PA1 2BE, UK

**Keywords:** heavy metals, contaminants, elements, toxic

## Abstract

Even if the Periodic Table of Chemical Elements is relatively well defined, some controversial terms are still in use. Indeed, the term “heavy metal” is a common term used for decades in the natural sciences, and even more in environmental sciences, particularly in studies of pollution impacts. As the use of the term appears to have increased, we highlight the relevance of the use of the term “Potentially Toxic Element(s)”, which needs more explicit endorsement, and we illustrate the chemical elements that need to be considered.

The development of the Periodic Table of Chemical Elements is one of the most significant achievements in science and a uniting scientific concept, with broad implications for the modern practice of chemistry, physics, biology and many other natural sciences. The International Year of the Periodic Table of Chemical Elements in 2019 coincides with the 150th anniversary of the discovery of the Periodic System by Mendeleev in 1869—a time to consider its application and progress to enable further exploration of our natural and human-impacted environment. It is a unique tool, enabling scientists to predict the appearance and properties of matter on Earth and in the Universe. However, even if the Periodic Table of Chemical Elements is relatively well defined, some controversial terms are still in use. Indeed, the term “heavy metal” is a common term used for decades in the sciences, and even more in environmental sciences (Figure 1), particularly in studies of pollution impacts. As the use of the term appears to have increased (annually 8% to 10% these last ten years, Figure 2), we emphasise the relevance of the use of the term “Potentially Toxic Element(s)” (PTEs) [1], which needs more explicit endorsement, and we illustrate the chemical elements that need to be considered.

In 1980, Nieboer and Richardson [2] had already proposed the replacement of this nondescript term by a biologically and chemically significant classification. Moreover, according to the International Union of Pure and Applied Chemistry [3], the term “heavy metal” is considered imprecise at best and meaningless and misleading at worst. The use of this term is strongly discouraged, especially as there is no standardised definition. In 2004, Hodson [4] considered them as geochemical “bogey men”. In 2007, Chapman [5] first proposed to keep this term for music not for science. In 2010, Hübner et al. [6] proposed to move on from semantics to pragmatics, whereas Madrid [7] recalls the long-standing and sometimes forgotten controversy. Nikimnaa and Schlenk [8] further insisted on the ill-defined term. In 2012, Chapman [9] continued to write on the cacophony not the symphony around “heavy metals” and Batley et al. [10] further presented a detailed discussion of its usefulness. However, some authors still proposed some form of a definition. In 2010, Appenroth defined them in plant sciences [11], and in 2018, Ali and Kahn [12] proposed their own “comprehensive” definition. More recently, Pourret and Bollinger [13] questioned the use of the term “heavy metals”—to use or not to use?—and Pourret [14] clearly proposed to ban this term from the scientific literature, but why?

Overall, the term “heavy metal” is based on categorization by density or molar mass (zinc or copper have relatively low density and molar mass compared to lanthanides and actinides). It is often used as a group name for metals (i.e., transition metals from vanadium to zinc) that are associated with contamination and potential toxicity. The “heavy metals” list is not clearly defined and often mixes metals, metalloids and non-metals without clear definition. Eventually, the pejorative connotation of “heavy” associated with the toxicity of metal induces a kind of fear in society. All so-called “heavy metals” and their compounds may have relatively high toxicity (e.g., lead or cadmium). Nonetheless, metals are not always toxic and some are in fact essential—depending on the dose and exposure levels and the receiving organism/population, the balance between essential or toxic may tip (e.g., nickel or zinc).

In this opinion presented to the *International Journal of Environmental Research and Public Health*, we look at progress in environmental sciences and medicine within a restricted sample population. Among the 167 articles with the “heavy metal” term in the title and the 996 with it in the subject, which we identified from a total of 12,700 articles published in *International Journal of Environmental Research and Public Health* in the widely used databases of Scopus and the Web of Science using the search term “heavy metal” (data accessed 10 October 2019), we found lead (Pb), cadmium (Cd) and zinc (Zn) to be the three most commonly studied elements (69%, 67% and 62%, respectively, when considering the term “heavy metal” in the title (*n* = 167), and 32%, 30% and 23%, respectively, when considering the term in the subject (*n* = 996); Table 1 and Figure 4). In addition, these elements are most often associated with monitoring based on total or extractable concentrations in soils, sediments or water, in order to characterise pollution, to perform risk assessment and to identify environmental exposure and health hazards (Figure 3). Apart from these chemical elements, the keyword “China” appeared then in 44% of the articles (73/167) and 58% of the articles were co-authored by researchers from Chinese institutions (97/167), reflecting in part the emergence of intense research activity on widespread environmental issues in the region. Also emerging reports in English language journals, perhaps has enhanced the growth of the term, a result of perpetuating the approach to an established and long-standing practice.

In environmental science, the chemical speciation (molecular form) of the elements is often overlooked [15]. The fact that chemical speciation is rarely considered may be because it is relatively expensive (time and resources) and inherently difficult to measure directly. Sometimes, fractionation analysis is performed, such as sequential chemical extraction, to identify the accessibility of portions of the total sample content. However, elements are mostly judged as toxic because of evidence relating to the toxicity of only a few of the chemical species in which they occur, often from laboratory-based acute exposure. Since their physical, chemical and biological characteristics depend on molecular structure and not its elemental constituents, so does its toxicity. Indeed, the toxicity of these PTEs, like lead and cadmium, depends on their speciation and concentration not only in a quantitative way but also qualitatively. Bioaccessibility and/or bioavailability should be considered. Overall, human exposure to lead by the addition of tetraethyl-lead to gasoline as an antiknock agent, or to lead paint, is well documented. However, the lead-acid battery does not pose a direct threat to humans through use but may generate environmentally hazardous waste [16].

Therefore, it is essential that environmental studies further consider the species present rather than the elemental constituent in order to create meaningful data. It thus becomes clearer that failure to properly consider the chemical speciation of elements can lead to poor risk assessment and bad use of legislation. Laws and regulations based on simple elemental analysis may wrongly consider environmental media or products as toxic and group them in the term “heavy metals”.

## Concluding Remark

To be consistent, researchers should only use well-accepted definitions. The series from V to Zn are considered the transition metals, As is a metalloid, Se a non-metal and Ba an alkali-earth (Figure 4). The best way to describe the studied elements is to name them clearly or to consider them as a group of elements (metals or metalloids) and, in the case of environmental studies published in journals like *International Journal of Environmental Research and Public Health*, to consider the term “Potentially Toxic Element(s)” (PTEs). The wider and less ambiguous term PTE can provide consistent and comparative use and more specialised definitions accepted only on the basis of a more refined characterisation. Some other terms, like “Potentially Toxic Analytes”, can also be applied. Eventually, one should continue to educate people to avoid the term “heavy metals”, especially in non-peer-reviewed regulations or governments’ research reports.

## Figures and Tables

**Figure 1 ijerph-16-04446-f001:**
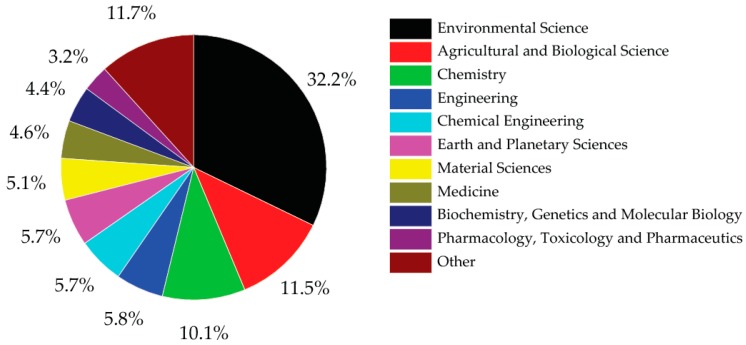
Proportion of publication by research areas in 2018 using the term “heavy metal*” in the title (sourced from Scopus using the term “heavy metal*”, data accessed on 10 October 2019).

**Figure 2 ijerph-16-04446-f002:**
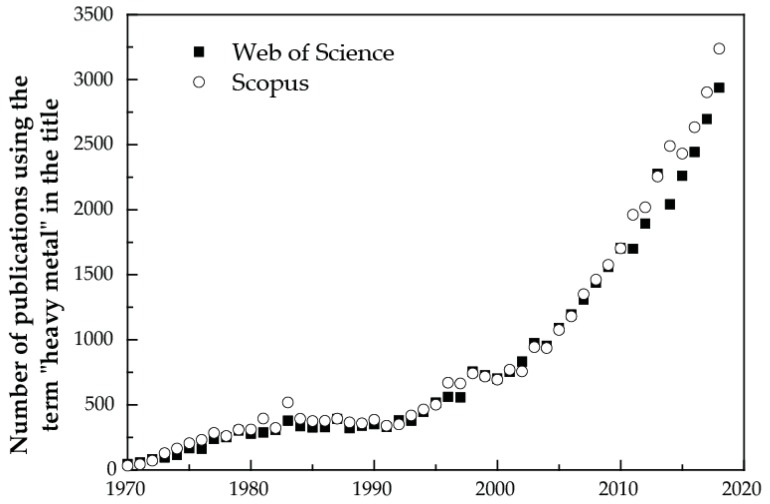
Evolution of the number of publications using the term “heavy metal*” in the title (sourced from Scopus and the Web of Science using the term “heavy metal*”, data accessed 10 October 2019). It should be noted that the total number of publications has also dramatically increased. Thus, the proportion of publications using this term may have decreased.

**Figure 3 ijerph-16-04446-f003:**
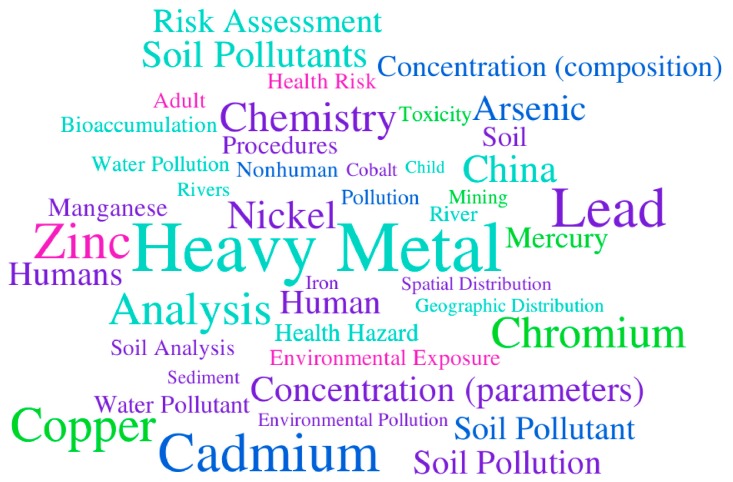
Word cloud of keywords used in the 167 articles from *International Journal of Environmental Research and Public Health* with the term “heavy metal*” in the title (sourced from Scopus, data accessed on 10 October 2019).

**Figure 4 ijerph-16-04446-f004:**
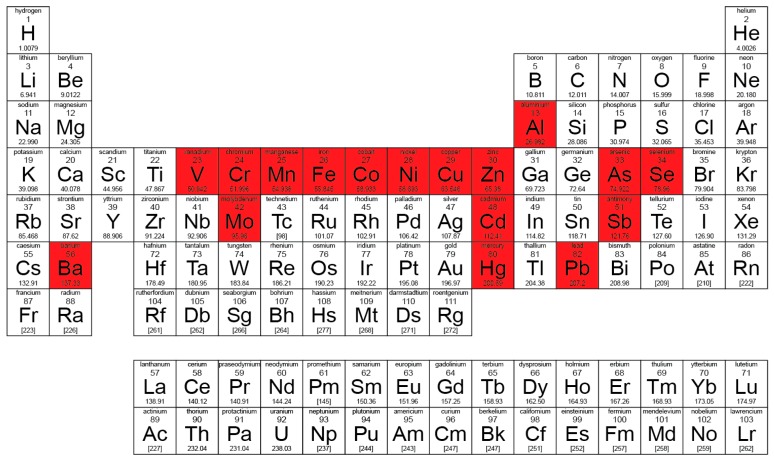
Periodic Table of Chemical Elements, highlighting the elements considered as “heavy metals” in *International Journal of Environmental Research and Public Health* (from the 167 articles with “heavy metal” in the title sourced from Scopus, data accessed on 10 October 2019).

**Table 1 ijerph-16-04446-t001:** Occurrence of the first 10 elements in article keywords with “heavy metal” or “potentially toxic element” in the subject of the articles published in *International Journal of Environmental Research and Public Health* (sourced from Scopus, data accessed on 10 October 2019).

Rank	“Heavy Metal” (*n* = 996)			Rank	“Potentially Toxic Element” (*n* = 131)		
	Element	*n*	%		Element	*n*	%
1	Pb	321	32	1	Pb	72	55
2	Cd	298	30	2	Cd	70	54
3	Zn	228	23	3	Zn	58	44
4	Cu	225	23	4	Cu	57	44
5	As	208	21	5	As	47	36
6	Cr	193	19	6	Cr	44	34
7	Ni	158	16	7	Ni	38	29
8	Hg	141	14	8	Hg	29	22
9	Mn	107	11	9	Mn	23	18
10	Fe	85	9	10	Co	20	15

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
