# Peer review of "It’s Time to Replace the Term “Heavy Metals” with “Potentially Toxic Elements” When Reporting Environmental Research"

_ijerph, 2019, doi:10.3390/ijerph16224446_

Round 1

Reviewer 1 Report

Original and interesting article in its approach.

Concise and appropriate article that will help a more precise use of common terms in the scientific literature of environmental issues. Interesting and well written.

Author Response

The authors sincerely thanks the reviewer for his/her sincere opinion. 

Reviewer 2 Report

The aim of this manuscript is clear that the authors aim to focus on replacing the term “heavy metals”. The title is informative and relevant. References are relevant and most recent. Although there are references back to the 1980s, the references are appropriate. The data was presented mostly in an appropriate way, but some improvements could be applied. It is clear that it is inappropriate to use the term “heavy metal”. However, I challenge the usage of “potentially toxic elements”. 

Major comments:

1. The authors mentioned the keyword “China” and Chinese institutions but did not indicate why this keyword was selected. Also, reference 14 was cited to speculate using this term as a traditional approach of long-standing practice. However, this reference did not mention any details regarding nationality nor this speculation.

2. This communication letter is discussed from multiple angles. However, the authors only selected one journal and one potential replacement for the term “heavy metals”. The conclusion of using “Potentially Toxic Elements” seems overinterpreted. Can we use “potentially toxic metals”, “potential toxic analytes” or simply “metals”?

Minor comments:

What IUPAC stand for? (L39) The authors stated that chemical speciation is often overlooked (L82-83). It should be noted that reference 15 is a non-peer reviewed version and self-citation. I suggest referring a peer-reviewed publication from another source. The authors justified the increasing use of this term by 8%-10% appearance in the last ten years (L29). It should be noted that the total number of publications in the last ten years might increase dramatically. Thus, the proportion of publications using this term might decrease.

Author Response

Reviewer #2

The aim of this manuscript is clear that the authors aim to focus on replacing the term “heavy metals”. The title is informative and relevant. References are relevant and most recent. Although there are references back to the 1980s, the references are appropriate. The data was presented mostly in an appropriate way, but some improvements could be applied. It is clear that it is inappropriate to use the term “heavy metal”. However, I challenge the usage of “potentially toxic elements”. 

The authors sincerely thanks the reviewer for his/her sincere opinion; according to his/her suggestions, we have rewritten some parts of the initial MS.

Major comments:

The authors mentioned the keyword “China” and Chinese institutions but did not indicate why this keyword was selected. Also, reference 14 was cited to speculate using this term as a traditional approach of long-standing practice. However, this reference did not mention any details regarding nationality nor this speculation.

The keywords was selected as one of the first one occurring (14th). Details were added. Moreover, reference was deleted.

This communication letter is discussed from multiple angles. However, the authors only selected one journal and one potential replacement for the term “heavy metals”. The conclusion of using “Potentially Toxic Elements” seems overinterpreted. Can we use “potentially toxic metals”, “potential toxic analytes” or simply “metals”?

Details were added. We only selected IJERPH as it is a letter, not a full article, for IJERPH. The same should be investigated in many other journals.

The conclusion was slightly modified. PT elements not metals should be used as PTE includes metalloids… Which is the problem of heavy metals…

 Minor comments:

What IUPAC stand for? (L39)

The acronym was now detailed.

The authors stated that chemical speciation is often overlooked (L82-83). It should be noted that reference 15 is a non-peer reviewed version and self-citation. I suggest referring a peer-reviewed publication from another source.

This paper is now in press, after peer-review, reference was modified accordingly.

The authors justified the increasing use of this term by 8%-10% appearance in the last ten years (L29). It should be noted that the total number of publications in the last ten years might increase dramatically. Thus, the proportion of publications using this term might decrease.

Thanks for this comment, details were added.

Reviewer 3 Report

This article is well-written and uses relevant data to support the points. I totally agree with the authors' points.  I do have two comments like to share.  

Lines 72-76:  It should be made clear that the statistic data are referring to the articles related to the 167 articles with the “heavy metal” term in the title.   It is not in the scope of this manuscript, however, it will be interesting to know how frequent the term "heavy metals" is used in regulations or governments' research reports.   

Author Response

This article is well-written and uses relevant data to support the points. I totally agree with the authors' points.  I do have two comments like to share.  

The authors sincerely thanks the reviewer for his/her sincere opinion; according to his/her suggestions, we have rewritten some part of the initial MS.

Lines 72-76:  It should be made clear that the statistic data are referring to the articles related to the 167 articles with the “heavy metal” term in the title.  

Details were added.

It is not in the scope of this manuscript, however, it will be interesting to know how frequent the term "heavy metals" is used in regulations or governments' research reports.   

Interesting comment but unfortunately we do not have any data on that. We have added a sentence in the conclusion.

Reviewer 4 Report

I agree this letter and the good explanation.

Author Response

(The authors gave the same response as above.)
